# Endosomally Localized RGLG-Type E3 RING-Finger Ligases Modulate Sorting of Ubiquitylation-Mimic PIN2

**DOI:** 10.3390/ijms23126767

**Published:** 2022-06-17

**Authors:** Katarzyna Retzer, Jeanette Moulinier-Anzola, Rebecca Lugsteiner, Nataliia Konstantinova, Maximilian Schwihla, Barbara Korbei, Christian Luschnig

**Affiliations:** Department of Applied Genetics and Cell Biology, Institute of Molecular Plant Biology, University of Natural Resources and Life Sciences, Muthgasse 18, 1190 Vienna, Austria; retzer@ueb.cas.cz (K.R.); jeanettemoulinier@boku.ac.at (J.M.-A.); rebecca.lugsteiner@boku.ac.at (R.L.); nataliia.konstantinova@psb.vib-ugent.be (N.K.); maximilian.schwihla@students.boku.ac.at (M.S.)

**Keywords:** ubiquitin E3 ligase, plasma membrane protein, protein stability

## Abstract

Intracellular sorting and the abundance of sessile plant plasma membrane proteins are imperative for sensing and responding to environmental inputs. A key determinant for inducing adjustments in protein localization and hence functionality is their reversible covalent modification by the small protein modifier ubiquitin, which is for example responsible for guiding proteins from the plasma membrane to endosomal compartments. This mode of membrane protein sorting control requires the catalytic activity of E3 ubiquitin ligases, amongst which members of the RING DOMAIN LIGASE (RGLG) family have been implicated in the formation of lysine 63-linked polyubiquitin chains, serving as a prime signal for endocytic vacuolar cargo sorting. Nevertheless, except from some indirect implications for such RGLG activity, no further evidence for their role in plasma membrane protein sorting has been provided so far. Here, by employing RGLG1 reporter proteins combined with assessment of plasma membrane protein localization in a *rglg1 rglg2* loss-of-function mutant, we demonstrate a role for RGLGs in cargo trafficking between plasma membrane and endosomal compartments. Specifically, our findings unveil a requirement for RGLG1 association with endosomal sorting compartments for fundamental aspects of plant morphogenesis, underlining a vital importance for ubiquitylation-controlled intracellular sorting processes.

## 1. Introduction

A strict regulation of protein homeostasis is of central importance for essentially every vital parameter in living organisms. Plants, with their sessile life style, in particular need to permanently respond to fluctuating environmental conditions, which involves flexible adjustments in protein levels and subcellular distribution [1,2]. A key role in controlling protein fate has been attributed to reversible substrate modifications catalyzed by E3 ubiquitin ligases. These enzymes are central to the attachment of the small protein modifier ubiquitin onto ε-amino groups of lysines, found in substrate proteins, which influences the half-life and/or subcellular localization of such target proteins [3,4].

Amongst the myriads of E3s identified in the genomes of higher plants, the so-called RING DOMAIN LIGASE (RGLG) RING finger E3 ligases appear especially relevant for the control of protein sorting, as they have been demonstrated to act in concert with specific protein co-factors, catalyzing the formation of ubiquitin chains on target proteins that are linked via lysine 63 (K63) of ubiquitin [5]. This type of substrate polyubiquitylation is of key importance for maintaining genome integrity via DNA repair mechanisms, but has also been intimately linked to the control of endocytic membrane protein sorting, followed by proteolytic turnover in the cell’s lytic compartments [6,7]. The *Arabidopsis* RGLG clade consists of 5 members, with RGLG2 demonstrated to function in K63-linked ubiquitin chain formation in vitro [5]. Furthermore, a combinatorial loss of redundantly acting *RGLG1* and *RGLG2* has been found to cause severe aberrations in plant morphogenesis, hormonal signaling and homeostasis, indicative of relevant roles for RGLG-catalyzed protein ubiquitylation in various aspects of plant development [5]. However, mechanisms by which RGLGs might impact on the control of membrane protein sorting and turnover, remained essentially elusive [7].

Strikingly, two members of the RGLG family, RGLG1 and RGLG5 have been demonstrated to control the protein half-life of PP2CA, a key negative regulator of abscisic acid (ABA) signaling in *Arabidopsis* [8]. This process likely involves K48-linked protein ubiquitylation and PP2CA sequestration for its degradation by the proteasome. Noteworthy in this context is that ABA has also been found to antagonize the myristoylation of RGLG1, resulting in its accumulation in the nucleus, where it is suggested to stimulate ubiquitylation of PP2CA [9]. A similar relocation to the nucleus has been described for mammalian RING proteins, albeit utilizing different mechanisms [10]. Additional evidence has been provided for RGLGs functioning in plant stress signaling and responses. RGLG2 was found to control proteasome-mediated turnover of AtERF53 transcriptional regulator, implicated in the control of drought-responsive genes [11]. RGLG3 and RGLG4, on the other hand, have been linked to Jasmonic Acid (JA) and Salicylic Acid (SA) signaling, and were attributed to modulate programmed cell death (PCD) [12,13]. More recently, RGLG1 and RGLG2 were demonstrated to catalyze in vitro K48-linked polyubiquitylation of MITOGEN ACTIVATED PROTEIN KINASE KINASE KINASE 18 (MAPKKK18), a key regulator of drought tolerance, further underlining the central function of RGLG proteins in the regulation of stress signaling in *Arabidopsis* [14].

In contrast, and although K63-linked poly-ubiquitylation has been demonstrated to play key roles in the control of endocytic sorting and vacuolar degradation of membrane proteins in higher plants, comparably little is known about the role RGLGs in these processes. Notable exception is the auxin transport protein PIN-FORMED 2 (PIN2), which undergoes K63-linked polyubiquitylation in dependence of functional RGLG1 and RGLG2, and exhibits increased protein steady-state levels in the *rglg1rglg2* double mutant [15]. Furthermore, a connection between iron deficiency responses and RGLG activity has been established, emphasizing links between K63-linked ubiquitylation of IRT1 iron transport protein and the activities of RGLGs and the like [16,17].

In this article, we describe an in-depth characterization of RGLG functions in endocytic cargo sorting in *Arabidopsis.* By employing compartment-specific marker lines and by determining the fate of selected intrinsic plasma membrane cargo proteins, we reveal a function for RGLGs in endosomal sorting decisions. This regulation seemingly involves the control of plasma membrane protein homeostasis by antagonizing protein targeting to the plasma membrane via secretory pathways.

## 2. Results

A clear-cut function for RGLGs has been demonstrated in the turnover of proteins residing in intracellular compartments, via proteasomal degradation [9]. Nevertheless, as there are indications for a role of RGLGs in K63-linked ubiquitylation, it is of great interest to study a function of RGLGs in membrane protein sorting via the endomembrane system. For this purpose, we first generated transgenic *Arabidopsis* lines, expressing an *RGLG1:Venus* fusion gene under control of an endogenous *RGLG1* promoter fragment. Expression of *RGLG1::RGLG1:Venus* in the *rglg1 rglg2* double mutant resulted in rescue of the mutant’s phenotypes, involving defects in apical dominance, late flowering and an overall semi-dwarf mutant stature [5], demonstrating that the reporter gene can functionally substitute for endogenous *RGLG1* and *RGLG2* (Figure 1A). We then viewed expression of the reporter gene and observed moderately intense fluorescent signals in the entire root meristem, exhibiting a rather diffuse distribution in the different cell files (Figure 1B). At a higher resolution, we detected disperse signals predominantly in the cytoplasm, together with small intracellular speckles, resembling vesicular structures in shape and size (Figure 1C, left panel). To learn more about these structures, we incubated *RGLG1::RGLG1:Venus* seedlings in presence of the endocytosed dye FM4-64 and tested for co-staining of RGLG1:Venus-positive structures (Figure 1C, middle and right panel). Indeed, this analysis revealed co-localization of FM4-64 and reporter protein signals, identifying these structures as endocytic vesicles and demonstrating that a subfraction of RGLG1:Venus locates to endosomal compartments. When comparing the kinetics of FM4-64 uptake in wild type and *rglg1 rglg2* root meristem cells, no striking differences could be observed, arguing against general deficiencies in the mutant’s endocytic sorting machinery (Appendix A), unlike the RING finger E3 ligase SALT TOLERANCE RING FINGER 1 (STRF1), which modulates endocytosis by altering expression of several genes involved in the membrane trafficking under salt stress [18]. Last of all, when determining the overall protein ubiquitylation status in cell lysates of the different plant lines, we did not detect obvious differences between wild type and *rglg1 rglg2* (Appendix A).

To learn about the subcellular localization of RGLG1:Venus signals in more detail, we made use of selected Wave marker lines [19], which we combined with *RGLG1::RGLG1:Venus* by crossing. A pronounced co-localization was observed between RGLG1:Venus and tagged Golgi structures-localized SYNTAXIN OF PLANTS 32 (SYP32:mCherry) [20] (Figure 2A). In addition, RGLG1:Venus shows co-localization with the presumptive secretory/recycling endosome marker RabA5d:mCherry (Figure 2B) and with late endosome/PVC marker RabF2b/ARA7:mCherry (Figure 2C). This positions RGLG1:Venus in compartments implicated in endocytic sorting to the vacuolar compartment as well as in recycling of plasma membrane-associated cargo via intracellular sorting vesicles. In contrast, only limited co-localization at the plasma membrane was observed, when co-expressing RGLG1:Venus with tagged plasma membrane marker plasma membrane intrinsic protein 1;4 (PIP1;4:mCherry; Figure 2D). Together, our experiments imply RGLG1 localization to defined populations of endosomal vesicular structures, pointing at a function in cargo sorting processes associated with these vesicles.

To further test the kinetics of RGLG1 localization in endosomal sorting compartments, we applied the fungal toxin Brefeldin A (BFA), which interferes with the function of ADP-ribosylation factor guanine-nucleotide exchange factors (ARF GEF) in the regulation of ARF activity in secretory vesicle budding [21]. As a result, protein cargo but also elements participating in the control of protein sorting and recycling pathways, accumulate in intracellular BFA-induced aggregations [22]. Indeed, BFA treatment resulted in pronounced accumulation of RabA5d:mCherry signals in such BFA compartments and a similar relocation was detected for RGLG1:Venus in response to the drug (Figure 3A). This BFA sensitivity of intracellular RGLG1:Venus distribution further underlines its association with intracellular sorting compartments, participating in the secretory pathway and/or in protein recycling between the plasma membrane and endosomal sorting compartments.

Published evidence provided so far is suggestive of RGLGs functioning in the control of soluble nuclear or cytoplasmic proteins, nevertheless our RGLG1 localization studies allow for deduction of a further working model. Therefore, and to obtain insights into the mechanistic significance of RGLG1 being associated with sorting vesicles, we tested consequences of drastically altered subcellular RGLG1 localization. RGLG1 and RGLG2 show a high degree of sequence identity and both share a protein myristoylation motif at the proteins’ N-termini [5], with protein myristoylation meanwhile demonstrated for RGLG1 and for RGLG2 [5,9]. Furthermore, and unlike wild type RGLG1, a myristoylation-deficient rglg1^G2A^ allele no longer accumulates in proximity of the plasma membrane but exhibited cytoplasmic and nuclear signals, when transiently expressed in tobacco leaf epidermis cells [9]. We made use of a similar approach, and generated stably transformed *rglg1 rglg2* lines expressing mutant *RGLG1::rglg1^G2A^:Venus*, which we compared with *rglg1 rglg2 RGLG1::RGLG1:Venus*. Unlike wild type RGLG1:Venus, rglg1^G2A^:Venus signals no longer accumulated in intracellular vesicular structures, but exhibited preferentially nuclear and cytoplasmic distribution (Figure 3B; top panels). We then tested consequences of short term BFA treatment, which in case of RGLG1:Venus results in pronounced signal accumulation in intracellular BFA compartments (Figure 3B; left bottom panel). Rglg1^G2A^:Venus in contrast, was no longer responsive to such treatment and retained its cytoplasmic and nuclear signal distribution (Figure 3B; right bottom panel). Such different subcellular distribution and sorting kinetics is consistent with scenarios in which loss of protein myristoylation prevents RGLG1 from entering BFA-sensitive vesicular sorting routes.

We then analyzed expression and steady-state protein levels in complementing *rglg1 rglg2 RGLG1::RGLG1:Venus* #4 and in *rglg1 rglg2 RGLG1::rglg1^G2A^:Venus* representative lines #5 and #54 on Western blots (Appendix A). Remarkably, in the *RGLG1::rglg1^G2A^:Venus* lines we observed a moderate shift in the migration of the reporter protein, potentially arising as a consequence of the mutant myristoylation site in rglg1^G2A^. We then assessed phenotypes of lines *rglg1 rglg2 RGLG1::rglg1^G2A^:Venus* #5 and #54, with #5 exhibiting reporter gene expression substantially more pronounced than in the *rglg1 rglg2 RGLG1::RGLG1:Venus* #4 control, and line #54 showing comparably poor reporter protein expression. Intriguingly, whilst *rglg1 rglg2 RGLG1::RGLG1:Venus* #4 plants exhibit phenotypes resembling wild type controls, *rglg1 rglg2 RGLG1::rglg1^G2A^:Venus* #5 and #54 failed to efficiently rescue defects associated with *rglg1 rglg2* plants at the stage of bolting and flowering (Figure 3C; Appendix A). These deficiencies are likely not a result of diminished expression of mutant rglg1^G2A^:Venus, indicated by the fact that even strong reporter protein overexpression in *rglg1 rglg2 RGLG1::rglg1^G2A^:Venus* #5 does not result in a rescue comparable to the situation in *rglg1 rglg2 RGLG1::RGLG1:Venus* #4, expressing wild type *RGLG1* (Figure 3C, Appendix A). Overall, this indicates that RGLG1 localization to endocytic sorting compartments is required for correct RGLG function in plant morphogenesis.

The partial rescue exhibited by strong overexpression in *rglg1 rglg2 RGLG1::rglg1^G2A^:Venus #5* demonstrates that the rglg1^G2A^ allele has still retained some of its functionality. We therefore asked, if this mutant allele shows activity in processes seemingly not linked to the control of plasma membrane protein sorting. Specifically, nuclear localization of RGLG1 was found previously to be induced by elevated ABA concentrations as a consequence of reduced protein myristoylation [9]. Elevated nuclear localization of RGLG1 in turn appears to promote degradation of PP2CA, thereby causing enhanced responsiveness to ABA [9]. We tested ABA responses of *rglg1 rglg2 RGLG1::RGLG1:Venus* #4 and *rglg1 rglg2 RGLG1::rglg1^G2A^:Venus* #5 in seed germination assays performed in presence of ABA. In these experiments, we found that *rglg1 rglg2 RGLG1::RGLG1:Venus* #4 seed germination is substantially less responsive to ABA when compared to the hyper-responsiveness of the *rglg1 rglg2 RGLG1::rglg1^G2A^:Venus* #5 seed germination, as reported previously (Appendix A) [9]. This indicates that myristoylation-deficient rglg1^G2A^, whilst no longer exhibiting a subcellular localization like wild type RGLG1:Venus, has retained functionality in modulating ABA responsiveness.

When taken together, comparison of wild type RGLG1 and rglg1^G2A^ expressing lines, demonstrated that efficient rescue of deficiencies in *rglg1 rglg2* morphogenesis coincides with RGLG1 localization to endosomal sorting vesicles. RGLG1 acting in such vesicles thus appears essential for proper *Arabidopsis* development, which links functions of these E3 ubiquitin ligase to the regulation of intracellular cargo sorting.

Our experiments are indicative of an essential role for RGLG1 in endosomal compartments, potentially acting in the control of protein ubiquitylation to influence the overall sorting of plasma membrane-originated cargo. The auxin export protein PIN-FORMED2 (PIN2) has been suggested to be controlled by RGLG activities, specified by increased abundance together with reduced K63-linked polyubiquitylation of PIN2 in *rglg1 rglg2* seedlings [15]. To obtain insights into such regulation, we determined localization of PIN2 and RGLG1 in *PIN2::PIN2:mCherry RGLG1::RGLG1:Venus* root meristem cells (Figure 4A). Expression of both markers exhibited some limited co-localization at the plasma membrane domain as well as in intracellular compartments. Furthermore, upon treatment with BFA we observed accumulation of both reporter protein signals in BFA-induced intracellular aggregations, indicating co-aggregation of RGLG1 and PIN2 in endosomal sorting compartments (Figure 4B).

Co-localization of marker proteins implies regulatory crosstalk, with a hypothetical function of RGLG1 in the regulation of PIN2 sorting and/or turnover. Analysis of *rglg1 rglg2 eir1-4* phenotypes is consistent with this working model, as this triple mutant exerts root gravitropism defects indistinguishable from *eir1-4* (a *PIN2* null allele [23]) single mutant phenotypes, whilst not displaying any additive phenotypes differing from those exhibited by parental *rglg1 rglg2* and *eir1-4* lines (Appendix A). To visualize effects of a loss of *RGLG1* and *RGLG2* on PIN2 protein fate, we then expressed *PIN2::PIN2:VEN* in *eir1-4 rglg1 rglg2* and compared its reporter protein expression with that of *eir1-4 PIN2::PIN2:VEN* (Figure 5A). No striking differences in PIN2 polar localization at the plasma membrane could be observed in *eir1-4 rglg1 rglg2*. Related results were obtained when determining abundance and localization of the plasma membrane markers PIN1:GFP and GFP-LTI6b. PIN1:GFP signals appeared unaffected in stele and endodermis root meristem cells and exhibited a basal plasma membrane localization in *rglg1 rglg2* seedlings, indistinguishable from wild type controls (Figure 5B). Similarly, GFP-LTI6b signals were found to localize apolarly to the plasma membrane in *rglg1 rglg2* root meristem epidermis cells, highly resembling its distribution in wild type controls (Figure 5C). Overall, the analysis of plasma membrane marker proteins in *rglg1 rglg2*, revealed no gross aberrations in subcellular localization, indicating that subcellular targeting of these plasma membrane proteins is essentially unaffected in the mutant.

In light of the apparently limited effects of *RGLG1* and *RGLG2* on plasma membrane protein sorting, we made use of another reporter line, suitable to detect and score very moderate effects on the fate of membrane proteins bound for endocytic sorting and degradation. *PIN2::PIN2:ubq:VEN* has been used earlier as a marker line to follow the trafficking of a constitutively ubiquitylated plasma membrane protein, which owing to a ubiquitin tag expressed in frame with PIN2, is efficiently sorted to and degraded in the vacuolar compartment [15,24]. Assessment of PIN2:ubq:VEN reporter signals in *eir1-4 rglg1 rglg2* root meristem cells revealed an increase in signal intensities when compared to *eir1-4 PIN2::PIN2:ubq:VEN* root meristem cells (Figure 6A). Strikingly, we detected signals at the plasma membrane as well as in the vacuolar compartments, especially in trichoblast cells, indicating that loss of *RGLG1* and *RGLG2* does not prevent PIN2:ubq:VEN from being sorted to the vacuole. However, when determining the ratio of signal intensities at the plasma membrane versus cytoplasm in *eir1-4 rglg1 rglg2 PIN2::PIN2:ubq:VEN* root meristem atrichoblast cells, we observed a significant increase in plasma membrane-bound signals, when compared to *eir1-4 PIN2::PIN2:ubq:VEN* (Figure 6A).

In conclusion, the increased build-up of reporter signals at the plasma membrane indicates a net decrease in the internalization of PIN2:ubq:VEN. The fact that there are still abundant signals detectable in the vacuolar compartments of *eir1-4 rglg1 rglg2 PIN2::PIN2:ubq:VEN*, however, demonstrates that vacuolar sorting is not completely blocked in this mutant. This is in stark contrast to the pin2^12K-R^ allele, which we analyzed in *eir1-4 rglg1 rglg2 PIN2::pin2^K12R^:VEN* root meristem cells (Figure 6B). This allele shows reduced ubiquitylation due to point mutations in potential ubiquitylation sites in the PIN2 central hydrophilic loop [15]. As a result of such ubiquitylation deficiency, the protein has been found to be retained at the plasma membrane in *eir1-4 PIN2::pin2^K12R^:VEN* and similar retention can be observed in *eir1-4 rglg1 rglg2 PIN2::pin2^K12R^:VEN* (Figure 6B). This indicates that RGLG1 and RGLG2 do not impact on the sorting and intracellular distribution of plasma membrane protein cargo that no longer gets efficiently ubiquitylated.

Overall, our observations led us to the conclusion that RGLG1 and RGLG2 impact on the sorting control of internalized ubiquitylated plasma membrane protein cargo in endosomal compartments. The stabilization specifically of ubiquitylation-mimic plasma membrane protein cargo upon loss of *RGLG1* and *RGLG2*, points towards a function of these E3 ubiquitin ligases in the cell’s communication with its environments via governing the sorting of cargo destined for proteolytic turnover.

## 3. Discussion

Plasma membrane protein fate is decisively influenced by intracellular sorting processes that define variations in overall subcellular protein distribution. In particular, adjustment of protein levels at the plasma membrane is of immanent importance for transport and signaling events that fine-tune intercellular communication in a tissue- or organ-dependent context [2,6,25,26,27]. Plasma membrane protein ubiquitylation represents a key event in aforementioned sorting decisions and findings presented here underline a function of RGLG1 E3 ubiquitin ligase in this particular process.

It is not entirely resolved, how substrate ubiquitylation defines subsequent adjustments in intracellular protein distribution. Experimental evidence led to working models, in which covalent attachment of even a single ubiquitin molecule appears sufficient for triggering downstream events that lead to endocytic internalization of protein cargo from its plasma membrane domains [6,28,29,30]. These events include the activity of adaptor proteins, recognizing (mono-)ubiquitylated cargo at the plasma membrane and guiding their sequestration into clathrin-coated vesicles for further sorting to endosomal compartments [24,31,32]. Once delivered, plasma membrane proteins face at least two choices: De-ubiquitylation, which would favor re-entry into secretory sorting routes, resulting in protein recycling to distinct plasma membrane domains [33,34]; or, alternatively further substrate decoration by K63-linked poly-ubiquitin chains, which would favor cargo transit into late endosomes or Multivesicular Bodies (MVBs), eventually resulting in cargo release into the lytic vacuolar compartment [30,35,36]. Together with the demonstrated enzymatic activity in K63-linked poly-ubiquitin chain formation for RGLG2 [5], the subcellular localization of closely related RGLG1 to endosomal sorting compartments is consistent with a function of these RGLGs in influencing the fate of endosome-located plasma membrane protein cargo. Specifically, RGLG-mediated poly-ubiquitylation occurring within such endosomal compartments could impact on further cargo sorting decisions.

The localization studies that we describe in this report place RGLG1 at defined intracellular BFA-sensitive endosomal sites, where the E3 could function in cargo sorting decisions between recycling and vacuolar degradation. This is underlined by the significance of a functional RGLG1 myristoylation domain, which appears essential for major aspects of RGLG1 activity in plant development. While the myristoylation-deficient *rglg1^G2A^* retains functionality in ABA responses, the inability of this allele to efficiently rescue the growth defects of the *rglg1 rglg2* double mutant favors the notion of functions of these E3 ubiquitin ligases associated with regulation of intracellular cargo sorting. This is consistent with the apparent stabilization of PIN2:ubq:VEN that we observed in *eir1-4 rglg1 rglg2* mutant background, presumably reflecting a shift in the cargo sorting equilibrium from vacuolar targeting to protein recycling to, or protein retention at the plasma membrane.

As judged from the subcellular localization of reporter signals, we found that vacuolar targeting of PIN2:ubq:VEN is not entirely blocked in *eir1-4 rglg1 rglg2*. Therefore, it is conceivable to conclude that these two E3 ubiquitin ligases are not solely responsible for mediating vacuolar turnover of the reporter protein. This argues for additional E3 ubiquitin ligases acting in the concerted decoration of plasma membrane cargo by K63-linked ubiquitin chains, next to the functionally characterized RGLG-type proteins. A number of candidates implicated in such enzymatic activities has been introduced in recent years. For example, IRT1 DEGRADATION FACTOR1 (IDF1) has been linked to the attachment of K63-linked poly-ubiquitin chains onto IRON-REGULATED TRANSPORTER1 (IRT1), thereby facilitating its vacuolar degradation [37,38]. In addition, PLANT U-BOX (PUB)-type E3s have been connected to the ubiquitylation and sorting control of different plasma membrane proteins, especially of brassinosteroid receptor protein BRASSINOSTEROID INSENSITIVE 1 (BRI1) [39,40,41]. Apart from that, interactome determination of UBC35 and UBC36, representing E2-conjugating enzymes rate-limiting for K63-linked polyubiquitin chain formation, revealed interaction with a diverse range of E3s, associating their activities to catalysis of K63 ubiquitin linkage [42]. The *Arabidopsis* RGLG clade consists of 5 members, with RGLG3 and RGLG4 found to form K48-linked polyubiquitin chains in vitro [12], and representing a subclade different from the RGLG1 and RGLG2 clade (Appendix A). RGLG5 appears phylogenetically closer to RGLG1 and RGLG2, potentially sharing a protein myristoylation site, but no confirmed null alleles have been characterized to date, hampering its functional analysis [8]. Thus, whilst *RGLG5* seemingly represents an essential locus, the biological function of this E3 awaits its further characterization.

A limited impact of RGLG1 and RGLG2 on plasma membrane protein sorting control is underlined by our analysis of reporter proteins in the corresponding double mutant. Wild type PIN2:VEN, PIN1:GFP and GFP-LTI6b all exhibited no detectable differences in subcellular distribution upon loss of *RGLG1* and *RGLG2*. Strikingly, whilst polar plasma membrane localization of PIN2:VEN appears unaffected, abundance of endogenous PIN2, has been demonstrated to be increased in *rglg1 rglg2*, which coincided with an evident decrease in its modification by K63-linked polyubiquitin chains [15]. These adjustments in PIN2 levels however, apparently do not compromise its function in the control of root gravitropism, as directional root growth appears unaffected in *rglg1 rglg2* [5]. Given the central, rate-limiting function for the control of plant morphogenesis and adaptation that has been attributed to members of the PIN protein family, this seems surprising on first sight. On the other hand, and presumably owing to their key roles, PIN functionality is subject to multifaceted regulation at transcriptional and foremost post-transcriptional levels [2,43,44]. The lack of *PIN2*-related phenotypes in *rglg1 rglg2*, thus could reflect this multitude of regulatory input, capable of compensating and balancing adjustments in PIN2 steady-state levels that might arise as a consequence of diminished K63-linked polyubiquitylation.

Nevertheless, fine-tuning of plasma membrane protein abundance, which we were only able to confirm with a reporter line with a very sensitive readout to changes in the endocytic degradation pathway, potentially does involves the function of RGLGs. We found that only wild type RGLG1:Venus is capable of fully rescuing major phenotypic traits that are associated with *rglg1 rglg2*. In contrast, the modified allele rglg1^G2A^ that no longer associated with endosomes, shows only partial rescue of *rglg1 rglg2* at the stage of flowering, but still exhibits functionality in mediating responses to ABA. *Rglg1 rglg2* mutant phenotypes concern leaf development, inflorescence formation, as well as control of flowering time, with our findings highlighting roles for endosome-associated RGLGs in plant morphogenesis. These *rglg1 rglg2* defects however do not coincide with generally altered endocytic sorting or an overall increase in the amount of ubiquitylated proteins, but appear of a more subtle nature, potentially affecting the fate of individual proteins Thus, and although PIN2 might represent only a subordinate target of RGLG activities, characterization of so far elusive substrates for RGLG-mediated K63-linked polyubiquitylation is likely to produce substantial insights into ubiquitylation-mediated membrane protein sorting control in higher plants.

## 4. Materials and Methods

### 4.1. Plant Lines, Growth Conditions and Vector Construction

Plants were grown on ½ x× Murashige Skoog (MS) medium (Duchefa Biochemie B.V; 2003 RV Haarlem, The Netherlands), or on PNS plant nutrient agar plates (5 mM KNO_3_, 2 mM MgSO_4_, 2 mM Ca(NO_3_)_2_, 250 mM KPO_4_, 70 μM H_3_BO_3_, 14 μM MnCl_2_, 500nM CuSO_4_, 1 μM ZnSO_4_, 200 nM Na_2_MoO_4_, 10 μM NaCl, 10 nM CoCl_2_, 50 μM FeSO_4_; pH adjusted to 5.7; supplemented with 1% (*w*/*v*) agar and 1% (*w*/*v*) sucrose; in a 16 hrs. light/8 hrs. dark regime at 22 °C). *PIN1::PIN1:GFP* [45], *GFP-LTI6b* [46], PIN2::PIN2:mCherry [47], *PIN2::PIN2:VEN* [15], *PIN2::PIN2:ubq:VEN* [15], *PIN2::pin2^12K-R^:VEN* [15]*, rglg1 rglg2* [5] and *eir1-4* [23] have been described elsewhere. Wave marker lines and GFP-LTI6b have been obtained from Nottingham Arabidopsis Stock Centre (NASC; https://arabidopsis.info/, accessed on 9 March 2022). Marker lines were introduced into *rglg1 rglg2 RGLG1::RGLG1:Venus* by crossing.

DNA-modifying enzymes were obtained from Thermo Fisher Scientific (https://www.thermofisher.com/at/en/home.html, accessed on 9 March 2022). For generation of *RGLG1::RGLG1:Venus* we first amplified a full-length *RGLG1* cDNA by using primers 5′-GGCCCGGGATGGGAGGAGGGAATTCCAAAG-3′ and 5′-CCGCCCGGGTCGACGTAAAGCTTGATTCTGGTCTGGATC-3′ as well as an *RGLG1* genomic promoter fragment by employing 5′-GGGAGCTCTTGATCAAGTCTTCTTATTCACACG-3′ and 5′-GGGAGCTCAAACTTTATTCAAATTAACAAAAAG-3′. The *RGLG1* cDNA was introduced into pPZPpApA [48] via *Cfr*9I, followed by addition of a 3′-end Venus tag via *Sal*I. The resulting *RGLG1:Venus* intermediate was then combined with the *RGLG1* promoter, which was introduced via *Sac*I to produce pRGLG1::RGLG1:Venus. This plasmid served as a template for site-directed mutagenesis by employing oligonucleotides 5′-GAGCTCGGTACCCGGGATGGCAGGAGGGAATTCCAAAGAAGAGTCGT-3′ and 5′-ACGACTCTTCTTTGGAATTCCCTCCTGCCATCCCGGGTACCGAGCTC-3′ resulting in the generation of pRGLG1::rglg1^G2A^:Venus, which was confirmed by DNA sequencing.

Flowering *Arabidopsis* plants were transformed with *RGLG1::RGLG1:Venus* and *RGLG1::rglg1^G2A^:Venus* by the floral dip method [49], using *Agrobacterium tumefaciens* strain GV3101/pMP90 [50]. Resulting T2 lines were confirmed for single transgene insertion sites and propagated to homozygosity for further analyses.

### 4.2. Microscopy and Data Acquisition

CLSM images were generated using a Leica SP5 (Leica Microsystems, Wetzlar, Germany) microscope. For imaging, we used the following excitation conditions: 488 nm (GFP); 514 nm (Venus); 561 nm (mCherry, FM4-64). For endocytotic sorting studies, 5–6-day-old seedlings were transferred from horizontally oriented nutrient plates into 6-well plates with liquid medium and incubated in presence of FM4-64 (Invitrogen, Thermo Fisher Scientific (https://www.thermofisher.com/at/en/home.html, accessed on 9 March 2022); working concentration 2 μM) for 10 min, followed by transfer into tap water for another 10 min before CLSM visualization. To follow up the kinetics of FM4-64 internalization, seedlings were incubated in presence of the dye for 10 min (working concentration 2 μM), followed by dark incubation in tap water for 15 and 60 min, respectively. BFA treatment was conducted according to [15].

For assessment of reporter protein distribution, we determined relative grey values at the plasma membrane and in endocytotic/vacuolar compartments, by using Fiji/ImageJ software [51]. Images acquired on a Leica SP5. For the quantification of CLSM fluorescence at the plasma membrane and in the cytoplasm, grey value intensity profiles and signal ratios were determined in at least three biological repeats.

### 4.3. Germination Assays

After surface sterilization of the seeds, stratification was conducted in the dark at 4 °C for 3 days. 100 seeds of each genotype were sown on ½ MS plates supplemented with 0.5 µM ABA or equal amounts of solvent as control. To score seed germination, radical emergence was analysed 48 h after placing plates in the incubator. Representative images were taken after 72 h. 

### 4.4. Statistical Evaluation and Sequence Alignments

All experiments were performed with at least three biological repeats. Statistical significance was calculated using one-way ANOVA with a post hoc Tukey HSD test. All images of control and chemical-treated samples were taken under the exact same settings of the microscope. RGLG protein sequences obtained from The Arabidopsis Information Resource (TAIR; https://www.arabidopsis.org/, accessed on 9 March 2022) were subject to multiple sequence alignments performed with ClustalX by using default settings [52]. The resulting alignment was used for determination of phylogenetic distances, by employing the FastME 2.1.6 tool [53]. Statistical support for branches was calculated with bootstrap replicas (*n* = 100).

### 4.5. Protein Analysis

Detection of global ubiquitylation and small-scale total protein extraction for western blotting were performed as described in [31]. SDS–PAGE, Coomassie brilliant blue staining, and immunoblotting were performed according to the standard protocols. Antibodies used in this study: Mouse-anti-ubq antibody (P4D1, Santa Cruz Biotechnology, Heidelberg, Germany; sc-8017); mouse-anti-GFP antibody (Roche, Basel, Switzerland; 11814460001); Goat anti-mouse immunoglobulin G (IgG) (horseradish peroxidase [HRP]-linked antibody; Jackson ImmunoResearch, Dianova, Hamburg, Germany; 115-035-164)

## Figures and Tables

**Figure 1 ijms-23-06767-f001:**
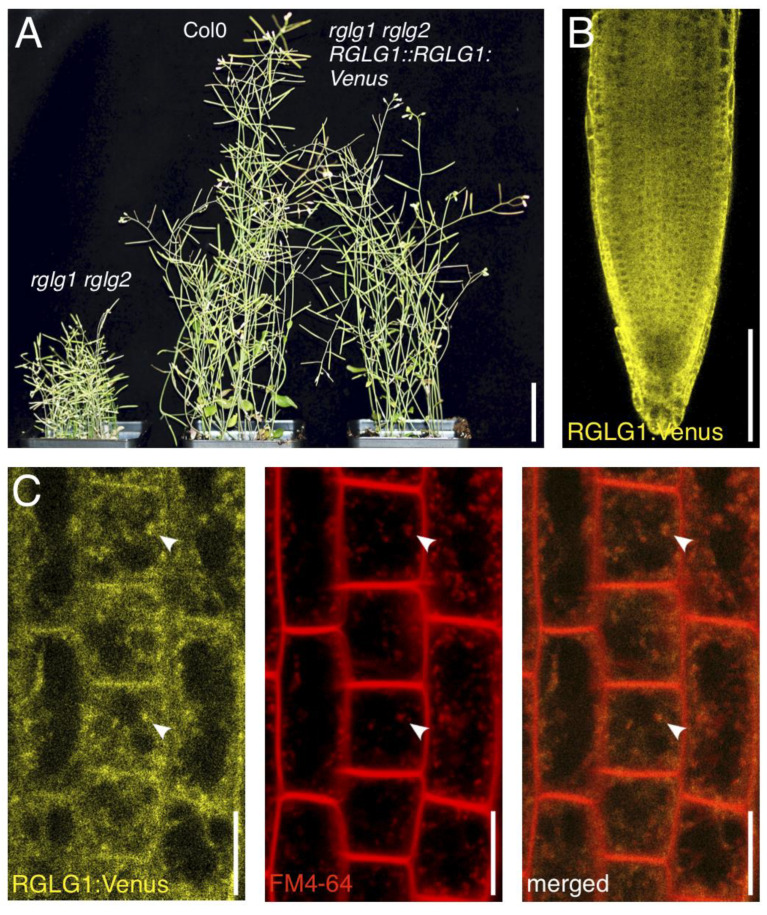
(**A**) Comparison of *rglg1 rglg2* (left; 48 days after germination) Col0 wild type (middle; 32 days after germination) and *rglg1 rglg2 RGLG1::RGLG1:Venus* (right; 32 days after germination) plants. (**B**) Expression pattern of *RGLG1::RGLG1::Venus* (yellow) in *rglg1 rglg2* roots at 6 days after germination. (**C**) Staining of *RGLG1::RGLG1::Venus* in *rglg1 rglg2* root meristem epidermis cells (yellow, left panel) with FM4-64 (red, middle panel) for 10 min in the dark, followed by visualization at the Confocal Laser Scanning Microscope (CLSM). White arrowheads indicate co-staining (orange, right panel) in endocytosed compartments. Size bars: (**A**) = 3 cm; (**B**) = 50 μm; (**C**) = 10 μm.

**Figure 2 ijms-23-06767-f002:**
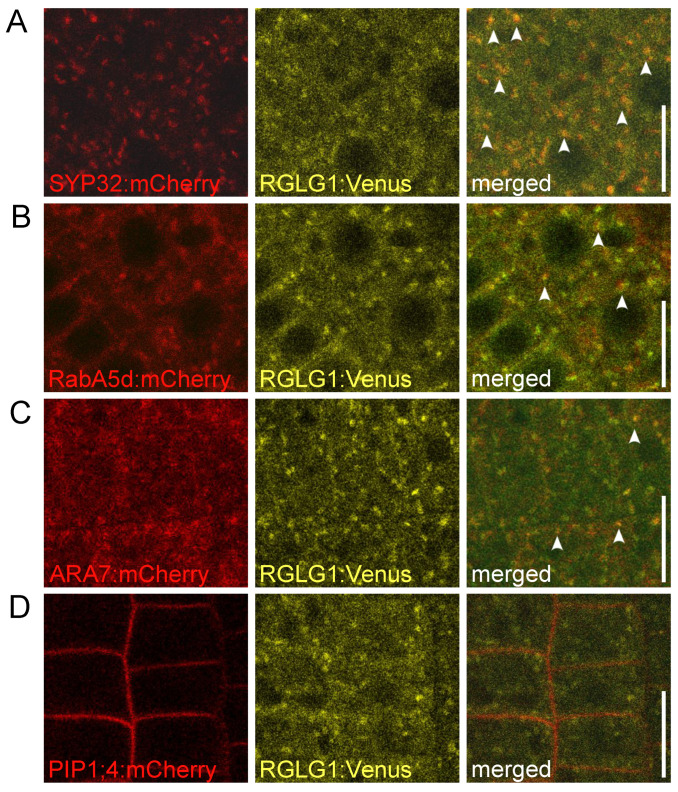
*RGLG1::RGLG1:Venus* (yellow) co-expressed with *UBQ10::SYP32:mCherry* (**A**; red) *UBQ10::RabA5d:mCherry* (**B**; red), *UBQ10::ARA7:mCherry* (**C**; red), and *UBQ10::PIP1;4:mCherry* (**D**; red). Images show root meristem epidermis cells at 6 days after germination. Arrowheads: co-localization of reporter signals. Size bars: 10 μm.

**Figure 3 ijms-23-06767-f003:**
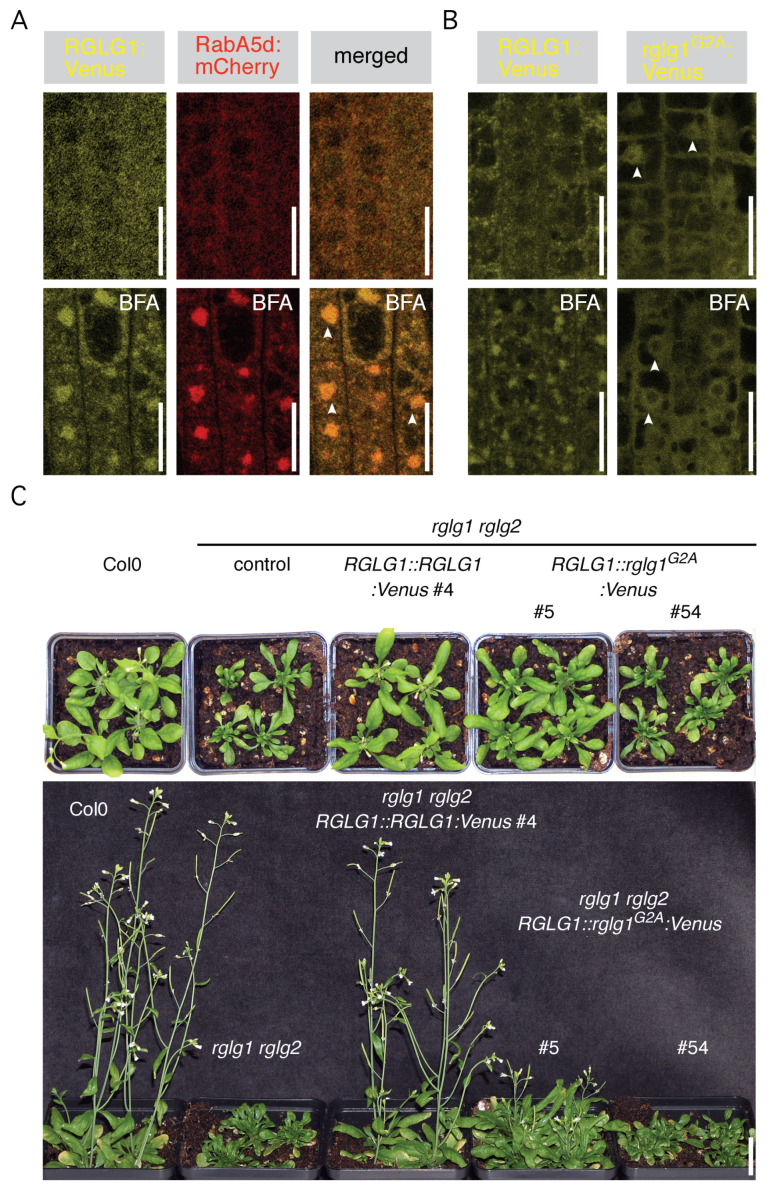
(**A**) RGLG1:Venus (yellow) and SYP32:mCherry (red) in root meristem mock-treated (top panels) or treated with BFA (50 μM for 90 min; bottom panels). Accumulation in BFA compartments: arrowheads. Size bars: 10 μm. (**B**) RGLG1:Venus (left panels) and rglg1^G2A^:VENUS (right panels) in root meristem mock-treated (top panels) or treated with BFA (50 μM for 90 min; bottom panels). Accumulation of rglg1^G2A^:Venus in nucleus: arrowheads. Size bars: 10 μm. (**C**) Top: Comparison of (from left to right) Col0, *rglg1 rglg2*, *rglg1 rglg2 RGLG1::RGLG1:Venus #4*, *rglg1 rglg2 RGLG1::rglg1^G2A^:Venus* #5 and #54 at the stage of bolting (24 days after germination). Bottom: Comparison of (from left to right) Col0, *rglg1 rglg2*, *rglg1 rglg2 RGLG1::RGLG1:Venus #4*, *rglg1 rglg2 RGLG1::rglg1^G2A^:Venus #5 and #54* at 32 days after germination. Size bar: 2 cm.

**Figure 4 ijms-23-06767-f004:**
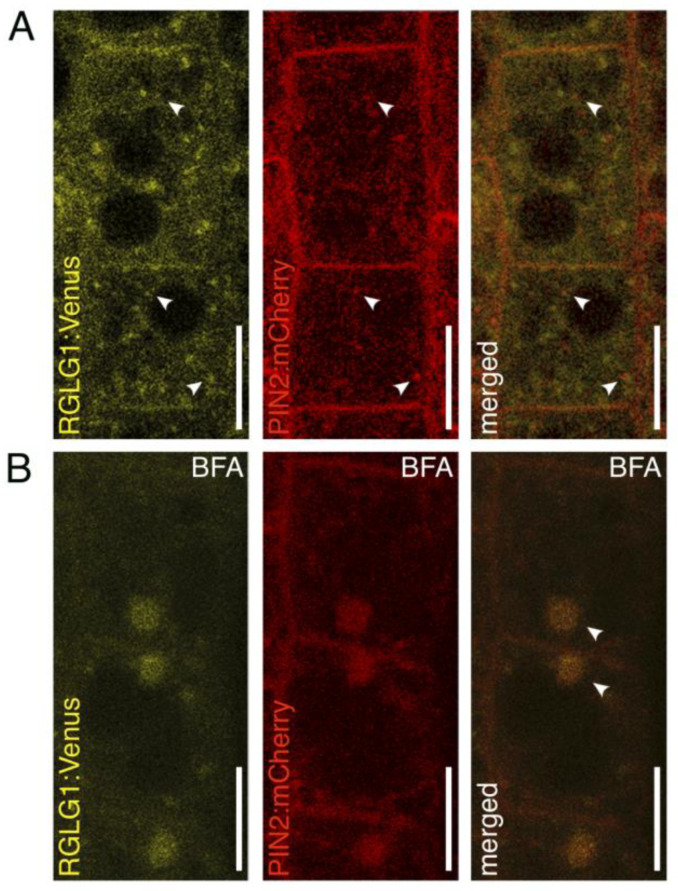
(**A**) RGLG1:Venus (yellow; left panels) and PIN2:mCherry (red; middle panels) in root meristem epidermis cells mock-treated. (**B**) RGLG1:Venus (yellow; left panels) and PIN2:mCherry (red; middle panels) in root meristem epidermis cells treated with BFA (50 μM for 90 min). Accumulation in intracellular compartments (‘**A**’; arrowheads) and in BFA compartments (‘**B**’; arrowheads). Size bars: 10 μm.

**Figure 5 ijms-23-06767-f005:**
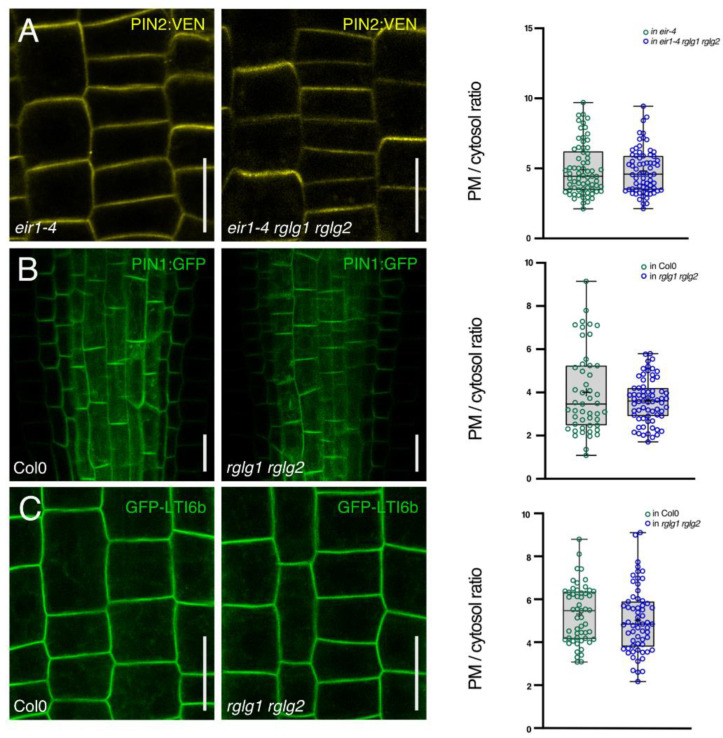
(**A**) PIN2:VEN (yellow) in *eir1-4* (left panel) or *eir1-4 rglg1 rglg2* (middle panel) root meristem cells at 5 days after germination. Right panel: Quantification of PIN2:VEN signals at the plasma membrane (PM) and in the cytosol in *eir1-4* or *eir1-4 rglg1 rglg2* root meristem epidermis cells (*n* = 65–70). (**B**) PIN1:GFP (green) in Col0 (left panel) or *rglg1 rglg2* (middle panel) root meristem stele cells at 5 days after germination. Right panel: Quantification of PIN1:GFP signals at the plasma membrane (PM) and in the cytosol in Col0 or *rglg1 rglg2* root meristem stele cells (*n* = 60–70). (**C**) GFP-LTI6b (green) in Col0 (left panel) or *rglg1 rglg2* (middle panel) root meristem epidermis cells at 5 days after germination. Right panel: Quantification of GFP-LTI6b signals at the plasma membrane (PM) and in the cytosol in Col0 or *rglg1 rglg2* root meristem epidermis cells (*n* = 60–70). Circles represent single data points; boxes: first and third quartiles; center line: median; ‘+’: mean value. Two-tailed *t*-test was employed to determine statistical significance. There was no significant difference found. Size bars: 10 μm.

**Figure 6 ijms-23-06767-f006:**
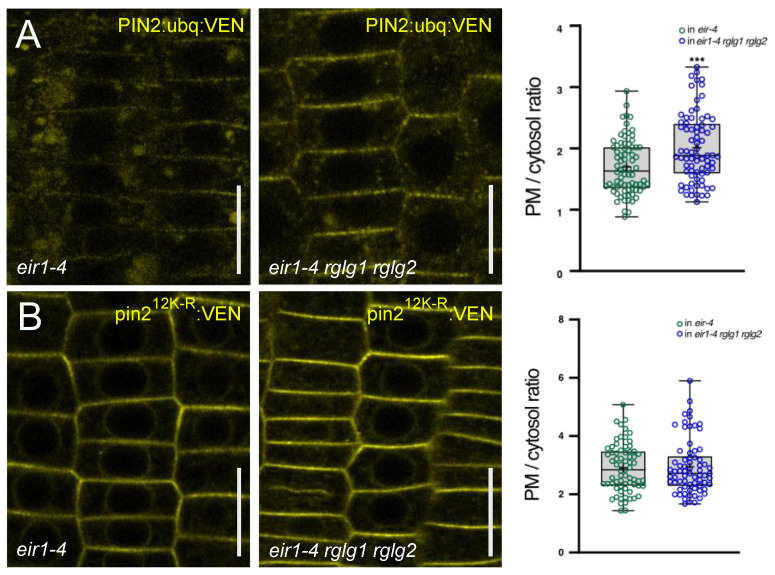
(**A**) PIN2:ubq:VEN (yellow) in *eir1-4* (left panel) or *eir1-4 rglg1 rglg2* (middle panel) root meristem epidermis cells at 5 days after germination. Right panel: Quantification of PIN2:ubq:VEN signals at the plasma membrane (PM) and in the cytosol in *eir1-4* or *eir1-4 rglg1 rglg2* root meristem epidermis cells (*n* = 70–80). (**B**) pin2^12K-R^:VEN (yellow) in *eir1-4* (left panel) or *eir1-4 rglg1 rglg2* (middle panel) root meristem epidermis cells at 5 days after germination. Right panel: Quantification of pin2^12K-R^:VEN signals at the plasma membrane (PM) and in the cytosol in *eir1-4* or *eir1-4 rglg1 rglg2* root meristem epidermis cells (*n* = 70–80). Circles represent single data points; boxes: first and third quartiles; center line: median; ‘+’: mean value. Two-tailed *t*-test was employed to determine statistical significance; ***: *p* < 0.001. Size bars: 10 μm.

## Data Availability

Data and materials generated by this study will be made available upon request, by the corresponding authors (B.K. and C.L.).

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
