# Peer review of "Endosomally Localized RGLG-Type E3 RING-Finger Ligases Modulate Sorting of Ubiquitylation-Mimic PIN2"

_ijms, 2022, doi:10.3390/ijms23126767_

Round 1

Reviewer 1 Report

In their manuscript, Retzer and colleagues sought to investigate the role of RGLG-type E3 RING-finger ubiquitin ligases in sorting ubiquitinated plasma membrane proteins in Arabidopsis thaliana. The manuscript is well written and pleasant to read.

However, the study focuses on RGLG1 – and RGLG2 to a lesser extent – and there is no direct evidence of ubiquitinated proteins being investigated in this study, making the title somewhat misleading. Hereafter are some additional comments

Main comments:

  1. Figure 1: endocytic assays should be performed not only in rescued roots but also in mutant roots. A time-lapse would also be beneficial to evaluate the dynamic of RGLG1 to be recruited to endocytic structures.
  2. The title of the manuscript highlights the sorting of ubiquitinated proteins. However, there is no evidence of the modulation of ubiquitinated proteins in the present manuscript. As such, the authors should revise their title accordingly or include some pieces of evidence related to plasma membrane protein ubiquitination (see below).
  3. Ubiquitination levels should be checked in mutants and rescue rglg1; is there any changes in the overall pool of ubiquitinated proteins? Or their localisation (plasma membrane vs endocytic compartments or cytosol)?
  4. Is the catalytic activity of RGLG1 important to its function? A catalytically inactive mutant (RGLG1H462Y) should be used for rescue experiments, as well as endocytic assays.
  5. What would be the phenotype of an RGLG2 rescue? Due to the high similarity to RGLG1, would it be identical, or would they still have different functions? It would be great to see such investigations. How is RGLG2 localised? Can RGLG2 expression rescue the phenotypes to the same extent as RGLG1? If the authors decide to focus exclusively on RGLG1, this should be more obvious in the title.
  6. Suppl Figure 1: wild-type and rglg1 rglg2 mutant without any rescue should also be shown. If there is a way to quantify the germination, this will be convenient.

Minor comments:

  1. Figure 3: Evaluation of expression level of the different constructs would be better assessed by RT-qPCR.
  2. Figure 4: the figure is not very sharp and appears blurry. Add a higher resolution figure before publication.
  3. The authors may consider ultrastructure analysis to evaluate the impact of rglg1 rglg2 on endocytic compartments.

Author Response

In their manuscript, Retzer and colleagues sought to investigate the role of RGLG-type E3 RING-finger ubiquitin ligases in sorting ubiquitinated plasma membrane proteins in Arabidopsis thaliana. The manuscript is well written and pleasant to read.

However, the study focuses on RGLG1 – and RGLG2 to a lesser extent – and there is no direct evidence of ubiquitinated proteins being investigated in this study, making the title somewhat misleading. Hereafter are some additional comments

Main comments:

  1. Figure 1: endocytic assays should be performed not only in rescued roots but also in mutant roots. A time- lapse would also be beneficial to evaluate the dynamic of RGLG1 to be recruited to endocytic structures.

>Authors' response:

Thank you very much for pointing this out! We have added the suggested FM4-64 uptake kinetics experiments in the revised manuscript (Supplementary Figure 1). No striking differences in the kinetics of FM4-64 signal internalization became apparent when comparing Col0 and rglg1 rglg2 root meristem cells, implying that general aspects of endocytic sorting are not strongly affected by this mutant.

  1. 2. The title of the manuscript highlights the sorting of ubiquitinated proteins. However, there is no evidence of the modulation of ubiquitinated proteins in the present manuscript. As such, the authors should revise their title accordingly or include some pieces of evidence related to plasma membrane protein ubiquitination (see below).

>Authors' response:

This point seems valid of course. Here we wanted to stress the fact that none of the plasma membrane proteins exhibited striking differences in rglg1 rglg2, except for a ubiquitylation-mimic version of PIN2. Whilst this points towards a role for RGLGs specifically in the sorting of ubiquitylated plasma membrane cargo, we tried not to generalize and changed the title of our manuscript accordingly.

  1. Ubiquitination levels should be checked in mutants and rescue rglg1; is there any changes in the overall pool of ubiquitinated proteins? Or their localisation (plasma membrane vs endocytic compartments or cytosol)?

>Authors' response:

We followed the reviewer's suggestion and analyzed global ubiquitylation in rglg1 rglg2 seedlings (Supplementary Figure 2 in the revised manuscript). These experiments revealed no difference in overall ubiquitylation in the mutant, when compared to wild type controls.

  1. Is the catalytic activity of RGLG1 important to its function? A catalytically inactive mutant (RGLG1H462Y) should be used for rescue experiments, as well as endocytic assays.

>Authors' response:

This of course would represent another interesting experiment, when it comes to functional analysis of RGLGs. For a proper experimental design, we would first need to ensure that the mutant protein no longer exhibits enzymatic activity in the formation of K63- and/or K48-linked ubiquitin chains. This typically requires protein expression in E. coli, followed by purification and setting up in vitro ubiquitylation assays, which in case of determination of K63-ubiquitin linkages is rather complex. Furthermore, there is of course no guarantee that such a drastic modification within the RING finger domain could affect overall conformation and hence stability of the mutant protein. All these potential pitfalls in experimental setup and subsequent analysis in heterologous hosts and in planta makes us believe that this type of experiment – whilst extremely interesting – is beyond the scope of the current manuscript.

  1. What would be the phenotype of an RGLG2 rescue? Due to the high similarity to RGLG1, would it be identical, or would they still have different functions? It would be great to see such investigations. How is RGLG2 localised? Can RGLG2 expression rescue the phenotypes to the same extent as RGLG1? If the authors decide to focus exclusively on RGLG1, this should be more obvious in the title.

>Authors' response:

Research published by Yin and colleagues (The Plant Cell; 2007; doi/10.1105/tpc.107.052035), also included an analysis of single T-DNA insertion lines, affecting either RGLG1 or RGLG2. Whilst these T-DNA lines have been demonstrated to impact on transcription of RGLG1 and RGLG2, respectively, which qualifies them as loss-of-function alleles, no phenotypic aberrations were observed in these single mutant lines. This argues for redundant functions of these two E3s and indicates that both genes perform identical functions. This is supported further by the complete rescue of rglg1 rglg2 phenotypes resulting from expression of 35S::RGLG1:GFP or 35S::RGLG2:GFP (Yin et al.; The Plant Cell; 2007; doi/10.1105/tpc.107.052035) as well as by RGLG1::RGLG1:Venus (this manuscript).

  1. Suppl Figure 1: wild-type and rglg1 rglg2 mutant without any rescue should also be shown. If there is a way to quantify the germination, this will be convenient.

>Authors' response:

We totally agree with the reviewer's objections. We therefore have added a statistical analysis of germination frequencies, further substantiating the differing responses of rglg1 rglg2 RGLG1::RGLG1:Venus and rglg1 rglg2 RGLG1::rglg1G2A:Venus to ABA treatment (Supplementary Figure 4).

Minor comments:

  1. Figure 3: Evaluation of expression level of the different constructs would be better assessed by RT-qPCR.

>Authors' response:

We have added the requested quantification, demonstrating that levels of the proteins are comparable (Supplementary Figure 3).

  1. Figure 4: the figure is not very sharp and appears blurry. Add a higher resolution figure before publication.

>Authors' response:

This must have happened upon conversion, for which we would like to apologize. We have replaced this version of Figure 4.

  1. The authors may consider ultrastructure analysis to evaluate the impact of rglg1 rglg2 on endocytic compartments.

>Authors' response:

We totally agree with this suggestion, as this might provide important insights into the function of the RGLGs. At this moment however, we do not have any strong evidence for more general sorting defects in the mutant (no aberrations observed, when determining endocytosis by FM4-64 as well as overall ubiquitylation status). We therefore have not yet initiated any experiments that would lead to establishment of such ultrastructure analysis in our labs.

Reviewer 2 Report

The manuscript, which title is “RGLG-type E3 RING-finger ubiquitin ligases function in ubiquitylated plasma membrane protein sorting”, is interesting. However, there are many studies have provided, such as Mani A, et al. The ubiquitin-proteasome pathway and its role in cancer. J Clin Oncol. 2005; Nobuhiro Nakamura The Role of the Transmembrane RING Finger Proteins in Cellular and Organelle Function. Membranes (Basel). 2011 Dec 9;1(4):354-93; Bocock et al. The PA-TM-RING protein RING finger protein 13 is an endosomal integral membrane E3 ubiquitin ligase whose RING finger domain is released to the cytoplasm by proteolysis. FEBS J. 2009 Apr;276(7):1860-77. Bocock et al. Nuclear targeting of an endosomal E3 ubiquitin ligase. Traffic. 2010 Jun;11(6):756-66. Tian et al. The RING finger E3 ligase STRF1 is involved in membrane trafficking and modulates salt-stress response in Arabidopsis thaliana. Plant J. 2015 Apr;82(1):81-92. Therefore the novelty of manuscript is limitation.

Author Response

The manuscript, which title is “RGLG-type E3 RING-finger ubiquitin ligases function in ubiquitylated plasma membrane protein sorting”, is interesting. However, there are many studies have provided, such as Mani A, et al. The ubiquitin-proteasome pathway and its role in cancer. J Clin Oncol. 2005; Nobuhiro Nakamura The Role of the Transmembrane RING Finger Proteins in Cellular and Organelle Function. Membranes (Basel). 2011 Dec 9;1(4):354-93; Bocock et al. The PA-TM-RING protein RING finger protein 13 is an endosomal integral membrane E3 ubiquitin ligase whose RING finger domain is released to the cytoplasm by proteolysis. FEBS J. 2009 Apr;276(7):1860-77. Bocock et al. Nuclear targeting of an endosomal E3 ubiquitin ligase. Traffic. 2010 Jun;11(6):756- 66. Tian et al. The RING finger E3 ligase STRF1 is involved in membrane trafficking and modulates salt-stress response in Arabidopsis thaliana. Plant J. 2015 Apr;82(1):81-92. Therefore the novelty of manuscript is limitation.

>Authors' response:

Thank you very much for these insightful comments! Of course, E3 ligases have already been implicated in the control of membrane protein sorting and have themselves being demonstrated to be under tight spatial regulation. However, for the most part these findings have been made in organismal classes different from higher plants. In particular, when it comes to the function of RGLGs in higher plants such as Arabidopsis, no clear-cut evidence for their role in the regulation of cargo sorting has been provided so far. Our manuscript presents such experimental evidence, which reveals a so far unknown function for this class of E3s.

We are very grateful for the references that have been provided by reviewer 2 and have added some of this literature in our revised version (Reference Nr. 10.      Bocock, J. P., S. Carmicle, E. Madamba, and A. H. Erickson. "Nuclear Targeting of an Endosomal E3 Ubiquitin Ligase." Traffic 11, no. 6 (2010): 756-66.n and Reference Nr. 18.Tian, M., L. Lou, L. Liu, F. Yu, Q. Zhao, H. Zhang, Y. Wu, S. Tang, R. Xia, B. Zhu, G. Serino, and Q. Xie. "The Ring Finger E3 Ligase Strf1 Is Involved in Membrane Trafficking and Modulates Salt-Stress Response in Arabidopsis Thaliana." The Plant journal : for cell and molecular biology 82, no. 1 (2015): 81-92.).

Reviewer 3 Report

The manuscript entitled “RGLG-type E3 RING-finger ubiquitin ligases function in ubiquitylated plasma membrane protein sorting “ focuses on RGLG1 sub-cellular localizations and intra-cellular functions, as well as its impact on Arabidopsis development. The overall merit of this study is particularly high considering the numerous transgenic plants that have been generated to thoroughly characterize the multiple localizations and functions of RGLG1/RGLG2. A few points need to be addressed to improve the quality of the manuscript, I think. Please find below my suggestions.

  • Figure 1 A: in order to evaluate the rescue level in rglg1 rglg2 RGLG1::RGLG1:Venus plants, it would be important to see the phenotype of WT plants as well.
  • Line 196-198 “In these experiments, we found that rglg1 rglg2 RGLG1::RGLG1:Venus seed germination is substantially less responsive to ABA when compared to the hyper-responsiveness of the rglg1 rglg2 RGLG1::rglg1G2A:Venus seed germination, as previously reported.” To support this conclusion, quantitative data should be shown with statistics highlighting significant differences in the germination rate (could be added to Supplementary Figure 1).
  • Better photos should be provided in the figure 4. It is hard to guess the co-localization of RGLG1:Venus and PIN2:mCherry in mock-treated roots.
  • Information about the eir1-4 mutant should be provided in the material and method section. The function of EIR1 should be explained as well.
  • Each abbreviation should be explained. For example, “CLSM” line 113.

Author Response

The manuscript entitled “RGLG-type E3 RING-finger ubiquitin ligases function in ubiquitylated plasma membrane protein sorting “ focuses on RGLG1 sub-cellular localizations and intra-cellular functions, as well as its impact on Arabidopsis development. The overall merit of this study is particularly high considering the numerous transgenic plants that have been generated to thoroughly characterize the multiple localizations and functions of RGLG1/RGLG2. A few points need to be addressed to improve the quality of the manuscript, I think. Please find below my suggestions.

Figure 1 A: in order to evaluate the rescue level in rglg1 rglg2 RGLG1::RGLG1:Venus plants, it would be important to see the phenotype of WT plants as well.

>Authors' response:

Thank you very much for pointing this out! We have changed Figure 1A accordingly.

Line 196-198 “In these experiments, we found that rglg1 rglg2 RGLG1::RGLG1:Venus seed germination is substantially less responsive to ABA when compared to the hyper-responsiveness of the rglg1 rglg2 RGLG1::rglg1G2A:Venus seed germination, as previously reported.” To support this conclusion, quantitative data should be shown with statistics highlighting significant differences in the germination rate (could be added to Supplementary Figure 1).

>Authors' response:

We fully agree with this suggestion! A statistical analysis of differences in seed germination has been added as requested, and can be found as Supplementary Figure 4.

Better photos should be provided in the figure 4. It is hard to guess the co-localization of RGLG1:Venus and PIN2:mCherry in mock-treated roots.

>Authors' response:

We have replaced the poor resolution version of this Figure. Marker localization should be more evident in the high resolution images used for Figure 4 in our revised version.

Information about the eir1-4 mutant should be provided in the material and method section. The function of EIR1 should be explained as well.

>Authors' response:

We have added the requested information in our revised manuscript.

Each abbreviation should be explained. For example, “CLSM” line 113.

>Authors' response:

We carefully went through the m/s, and explained all the abbreviations used throughout the revised manuscript.

Round 2

Reviewer 1 Report

Retzer and colleagues have provided extended and sensible answers to my previous comments in their revised manuscript and associated responses. They have amended their manuscript and added new data sets that reinforce their conclusions.

Only one comment on my side remains:

Supplementary Figure 3: Thank you for checking the expression level of the rescue constructs. There seems to be a shift in the protein size of the expressed constructs (G2A mutant appears at a lower MW than WT RGLG1. Is this due to the lack of myristoylation? It also seems that there is quite some difference in the expression level between the 2 proteins. Is G2A mutant somewhat more stable than WT?

Author Response

Retzer and colleagues have provided extended and sensible answers to my previous comments in their revised manuscript and associated responses. They have amended their manuscript and added new data sets that reinforce their conclusions.

Only one comment on my side remains:

Supplementary Figure 3: Thank you for checking the expression level of the rescue constructs. There seems to be a shift in the protein size of the expressed constructs (G2A mutant appears at a lower MW than WT RGLG1. Is this due to the lack of myristoylation? It also seems that there is quite some difference in the expression level between the 2 proteins. Is G2A mutant somewhat more stable than WT?

>Authors' response

Thank you very much for going through our revised manuscript!

Yes indeed, we noticed a shift in the migration of mutant rglg1G2A:Venus, when compared to the wild type reporter. Loss of myristoylation, most likely cannot account for this difference in protein migration. However, protein myristolyation might function as a trigger for further post-translational modifications, such as protein phosphorylation or further lipid modifications. Deficiencies in such modifications as a result of loss of RGLG1 myristolyation might offer an explanation for the apparent difference in protein migration, and, as reviewer #1 rightfully brought up, could also affect stability of the mutant protein.

Further experiments would be required to address these issues in sufficient detail.

Reviewer 2 Report

None

Author Response

>Authors' response

Thank you very much for going through our revised manuscript!